# Sustainable Assignment of Egyptian Ornamental Stones for Interior and Exterior Building Finishes Using the AHP-TOPSIS Technique

Ahmed M. A. Shohda [1,2], Mahrous A. M. Ali [2], Gaofeng Ren [1,3,*], Jong-Gwan Kim [4], Ahmed M. Abdo [5], Wael R. Abdellah [6] and Abbas M. Hassan [5]

[1] School of Resources and Environmental Engineering, Wuhan University of Technology, Luoshi Road 122, Wuhan 430070, China; shohdaahmed@azhar.edu.eg
[2] Mining and Petroleum Engineering Department, Faculty of Engineering-Qena, Al-Azhar University, Cairo 83513, Egypt; mahrousali@azhar.edu.eg
[3] Key Laboratory of Mineral Resources Processing and Environment of Hubei Province, Luoshi Road 122, Wuhan 430070, China
[4] Department of Energy and Resources Engineering, Chonnam National University, Gwangju 61186, Korea; kimjg@jnu.ac.kr
[5] Department of Architecture, Faculty of Engineering, Al-Azhar University, Qena 83513, Egypt; ahmedabdo.3821@azhar.edu.eg (A.M.A.); am.hassan@azhar.edu.eg (A.M.H.)
[6] Department of Mining and Metallurgical Engineering, Faculty of Engineering, University of Assiut, Assiut 71515, Egypt; waelabdellah@aun.edu.eg
[*] Correspondence: rgfwhut@163.com or rengf110@whut.edu.cn; Tel.: +86-18986194926

**Abstract:** The ornamental stones industry has grown progressively in Egypt as the demand has increased for Egyptian decorative stones in indoor and outdoor building spaces. Choosing the most suitable ornamental stone for each purpose and taking the right decisions related to aesthetic and practical performance can be a challenge causing a lot of confusion for homeowners and contractors. Thus, there is a need to define what decorative style you are trying to achieve in order to properly choose the most suitable decorative stone. In this research, some Egyptian ornamental stones were evaluated by combining the analytic hierarchy process (AHP) and the technique for order preference by similarity to ideal solution (TOPSIS). The optimized AHP–TOPSIS comprehensive decision model was implemented on natural materials relevant to ornaments and the finishing purposes of indoor and outdoor buildings. Ten rock types from four Egyptian sites were studied, examined, and analyzed. According to the quality index scheme, grey granite is an ideal ornamental stone that meets indoor and outdoor purposes. Grey granite and black marble have a performance score (Pi) of 0.78 and 0.71, respectively. Serpentine and pink granite were ranked third with a Pi = 0.68. These results provide stakeholders with strategic indicators to select different natural ornamental stones based on the total points assigned to all rock specifications and costs.

**Keywords:** TOPSIS; AHP; ornamental stone; quality index; marble; granite; UCS

## 1. Introduction

Since ancient times, man has been using natural stones as premium construction materials. After the propagation of cement–concrete-based materials, markets experienced a slowdown in the use of natural stones, but with recent improvements in quarrying, cutting, and supplying technologies, natural building stones have returned to be used for various objectives in the construction industry (coating, walls, pavements, flooring, etc.). Ornamental stones are stones that are used for a decorative effect. The rock of this stone has beauty, durability, and stability. Therefore, it is mined to be used in the construction of monuments [1]. Herein this study sheds light on the problem that stands in the way of selecting the most suitable ornamental stone and the practical relevance of their study.

To begin with, selecting the appropriate stone for each purpose and making the right decision regarding its workability and employability can be difficult and confusing for homeowners and contractors, as well as meeting all the usual aesthetic criteria such as color, pattern, texture, and so on. The stone must have good resistance, durability, and slip resistance [2]. Furthermore, geographical proximity, ease of mining, and transportation of the ornamental rocks as blocks from the quarry make the selection process of one of those stones in indoor and outdoor building spaces a challenging issue.

Second, numerous researchers have studied natural constructing stones and evaluated their mechanical and physical properties [3–13], but the majority of these studies did not focus on the integration between all physical and mechanical properties and the cost as an essential parameter for evaluation, which is considered as another challenge and causes homeowners' decisions to be hesitant, especially when these factors relate to the aesthetic criteria.

Third, as a part of operational research, Multi-Criteria Decision-Making (MCDM) methods are gaining popularity as a tool for studying and solving complex problems due to their inherent ability to evaluate exceptional alternatives in relation to various criteria for feasible selection of the best alternative [14]. Selecting the proper Decision-Making Model is a crucial task. Each approach has strengths and weaknesses; while a few techniques are higher grounded in mathematical theory, others can be easier to implement [15]. Therefore, our study presents one of the considerable methods of MCDM related to ornamental stones [15].

In terms of selecting a particular approach of MCDM, it is vital to recall the complexity of the selection in phrases of scientific, technical, and social factors, in addition to understanding the desired manner and the provision of information and/or understanding, approximately, regarding the problem space [16]. Several MCDM methods have been applied for solving problems in the areas of sustainable engineering [17–25]. Stojcic et al. [26] carried out a review paper regarding the application of MCDM strategies within the field of sustainable engineering during the period 2008–2018. After reviewing 329 articles from the Web of Science Core Collection database, it was observed that AHP was the most frequently used method among other MCDM methods in this area. Of the several MCDM techniques, the combination of AHP and TOPSIS techniques has been used. The justification for hybridizing these two MCDM techniques was based on each one's peculiar characteristics, and these features were uniquely used at different phases/stages of the computational evaluation. In addition to AHP's extensive use within and outside the mining industry, its ability to model complex decision problems makes it ideal [27,28]. The TOPSIS technique is a simple, rational, comprehensible concept with intuitive and clear logic that represents the rationale of human choice, ease of computation, and good computational efficiency; it is a scalar value that accounts for both the best and worst alternative abilities to measure the relative performance for each alternative in a simple mathematical form, with a possibility for visualization [29,30].

Egypt is one of the largest producers of ornamental stones, (Egypt, China, Italy, India, and Spain). Among 37 countries, these five countries contribute more than two-thirds of the 53 million metric tons of total global production. Egypt produces more than 10 types of ornamental stones, such as those made of grey granite, pink granite, marble, basalt, breccia, and serpentinite [31,32]. Granite rocks are extensively dispensed all around the Egyptian shield, constituting about 60% of the plutonic assemblage. Their hues are frequently white, pink, rose, grey, red, black, and their derivatives. They vary in composition from quartz diorite and tonalite to granodiorite and quartz monzonite to ordinary granites and alkaline–peralkaline granites [33,34]. Marble is a precious ornamental stone used by man for a long time. Real marble is found within the basement rock. It is a crystalline limestone that is characterized by diverse properties of dolomite. This kind of marble is available mainly in two places in Egypt. The first place is Wadi Al Miyah, which is located in the Eastern desert between Edfu and Marsa Alam. Here, natural marble has different colors, mainly

white, black, and grey. The second place is El sheikh Fadel, which is in the southeast of Ras Ghareb.

Serpentinite is a metamorphic rock used by ancient Egyptians as a building material and ornamental stone. Serpentinites are commercially known as Green Marble. However, this name has no correspondence to the actual mineralogy, geochemistry, and/or physical properties of serpentinites. Serpentinites are widely distributed in the Eastern Egyptian desert, particularly in the central and southern parts. Owing to the visual appearance and good geochemical properties of serpentinites, they are still an excellent choice as ornamental stones, which are mainly used for constructional purposes [35–37].

In this study, we aim to evaluate some Egyptian ornamental stones for different purposes (indoor and outdoor use) by adopting the technique for order preference by similarity to ideal solution (TOPSIS). The rank is assigned according to the integration between rock properties. This technique considers the weight (related to the analytic hierarchy process AHP technique) and rate of rock properties relevant to different purposes.

The rest of this paper is prepared as follows. In Section 2, the decorative stones utilized in indoor and outside areas are described. In Section 3, the vicinity and geological settings are described. In Section 4, laboratory test results of the mechanical and physical properties of rock samples are summarized. The literature evaluation of the AHP approach is mentioned in Section 5, wherein it suggests how we estimate the weight of stones. In Section 6, the TOPSIS approach and the utility of TOPSIS is mentioned. The proposed approach of assessment is mentioned in Section 7. Section 8 gives the results and discussions. Section 9 gives the limitations and implications of the study. Finally, the conclusions of this study are provided in Section 10.

## 2. Ornamental Stones in Indoor and Outdoor Building Spaces

In 1828, Heinrich Hubsch published a book titled "In which style should we build?". His question was echoed later by the leading neoclassicist Karl Friedrich Schinkel: Every major period has left behind its own style of architecture. Why should we not find the style for ourselves? Thus, modernists advocated the spirit of the age (Zeitgeist). Regarding classic building materials, such as natural stones, the neo-classic trend is still attractive among designers and habitats. The combination of glass and natural stones on outdoor or indoor walls gives a wonderful contrast. For example, a custom-designed base with conventional legs, a pedestal base, or an ornamental base utilizing different materials, together with wood or steel, is a possibility to convey extra materials, and an ornamental pedestal base manufactured from natural stone can be mixed with a glass tabletop and included into the design. The intervention among modern and conventional finishing materials could be applied within the floors of residential or industrial settings [38]. Natural stones used as ornaments are still desirable for their gorgeous appearance and durability.

Natural stones have been widely used as interior and exterior finishing materials. The sustainability of these stones makes them qualified to be among the top choices of architects when designing and building projects. The foyer is the primary influence of any building. The first few steps into the area set the tone for the relaxation experience at home, in the office, or a hotel. Selecting the material accurately could increase the value of a prestigious hall or entrance. Granite is a sustainable floor material as it resists erosion. Moreover, it is extraordinarily long lasting at the same time as offering aesthetics. Marble is another foyer option. Although it is more beautiful than granite, it is inferior to granite from a sustainability perspective. Some marble stones have pores on their surface, making them inappropriate for outdoor flooring spaces. Porous materials can be difficult to maintain. Thus, the use of marble in the main entrances of buildings is questionable, especially when people entering the building carry rain, snow, mud, etc., with them [39]. However, natural stones have additional uses.

The most common uses of natural stone are kitchen and bathroom countertops. However, these natural stones can be used in many other places to meet the aesthetics of the constructed environment. In any urban area, natural stones forming entrances, pedestals,

outdoor benches, etc., can be found. They are also used as basins, altars, and ornate fixtures in churches. Moreover, casinos or hotels usually have a marble floor, and outdoor kitchens generally use granite as the cladding material [40].

The previous examples are some alternative uses for natural stones in the urban environment. The implementation of natural stones exceeded the traditional use for more creative and innovative purposes [41,42]. The sophisticated use of natural stones has been a challenge for architects to use them in novel purposes. Natural stone is a beautiful alternative to hardwood, tile, and masonry (in some cases). Granite and marble are extraordinarily long lasting as they resist time and weather extrusions in outdoor finishes. In terms of their cost, the final fee of natural stones can be considered to be affordable in comparison to different comparable construction materials, including wood, due to the fact that the running fee of maintenance is nearly negligible. The flexible availability of natural stones in furnishings stores is taken into consideration as a further positive in comparison to different materials. From a modern and glossy dining room table to coffee tables or outdoor furnishings, natural stones are definitely the essential implemented material. Natural stones are without doubt plausible in indoor and outdoor furnishings. For instance, tablets may be bought as ready-made tables or made consistent with the use of a stone fabricator [43]. Moreover, natural stones are considered the ideal material for fireplaces.

Fireplaces and human dwellings were inseparable in most climates until recently. The fireplace and means of cooking have been seen by many as the beginning of architecture. Few modern homes have open fireplaces owing to the convenience of central heating and regulations on emissions. However, gas fires and wood burners are still valued as a means of offering the ambiance of a real fire. Natural stones, including granite, marbles, and different types of stacked stones, are sustainably used to construct indoor fireplaces, which meet various tastes [44]. Figure 1 shows the appearance and uses of some ornamental stones.

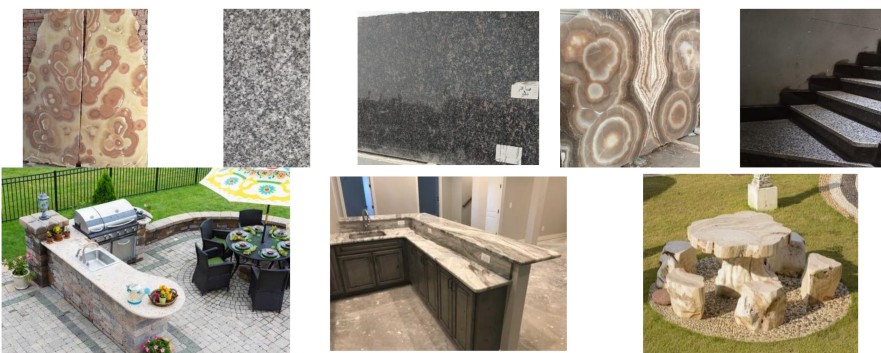

**Figure 1.** Ornamental stones used indoors and outdoors as decorative materials.

## 3. Location and Geological Settings

In this study, we investigated three samples of black marble, two samples of white marble, sunny marble, grey granite, pink granite, red granite, and serpentine. All samples were accrued from the study region illustrated in Figure 2. Wadi El-Myah is a crucial wadi in Egypt. The rock specification changed massively along with being at a moderately higher elevation from the ground of the wadi. The extracted samples ranged from gray to gray black in color and were dissected via white veins. The second region was El Barramia, in which the white marble is commercially called Carrara. This marble was white and gray in color with massive fine grains. Sunny marble is observed in El Shikh Fadel marble quarry. This marble is white to brownish in color, fine-grained, and consists of a few lenses of quartz with a few fractures. It consists frequently of calcite with a few traces of quartz and iron oxides. This rock seldom consists of fossils [45].

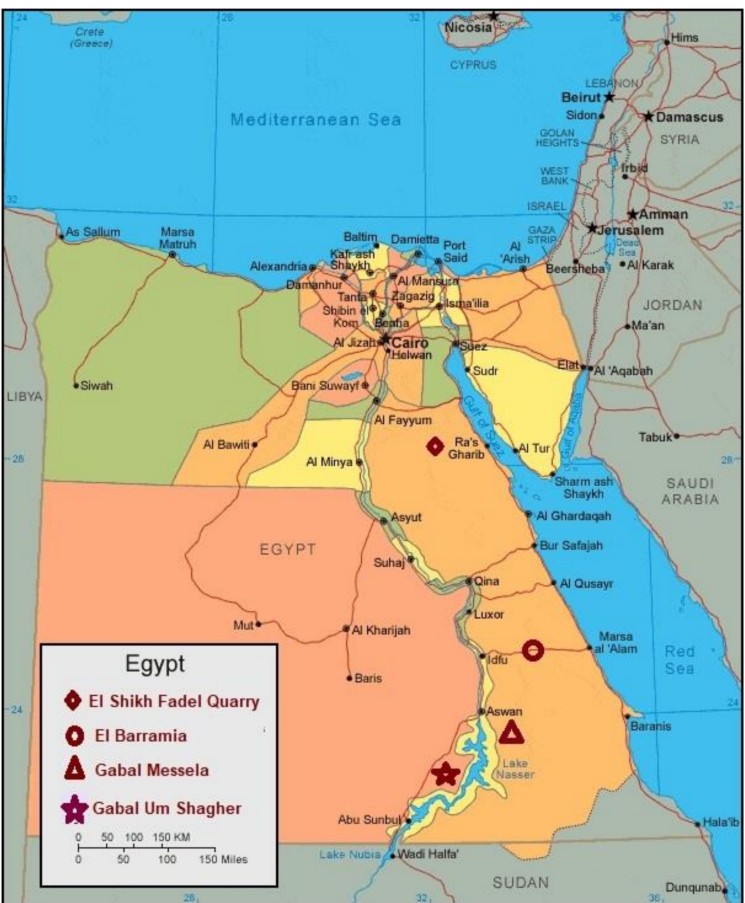

**Figure 2.** Location map of the study area.

Three different sorts of granite from three quarries are offered in this study as follows: (1) Grey granite, that is vintage granite, is observed in the El Barramia region shown in Figure 2. It is fine- to medium -grained and massive. It is characterized by quartz veins and joints. It consists of quartz, plagioclase, biotite, microcline, and different accent minerals. Quartz is colorless with a euhedral shape.

(2) Red granite is observed in the Gebel Um Shagher region (Figure 2). It is a younger granite that has comparable characteristics, such as morphology, color, mineralogical composition, and crystal size, to the well-known Aswan red granite. It is coarse- to very coarse-grained and has a subdural granular texture.

(3) Pink granite is observed in the Gabal El-Messala region (Figure 2). It is a younger granite characterized by a semicircular, coarse grain, and pink color. This region is the best area for Nubian sandstone and basement rock (granite). Large plutons and large blocks signify these granitic rocks. Hence, it is used economically as a constructing material [46,47].

Serpentine is observed in the Barramia region (Figure 2). It is called antigorite, which is a vital serpentine mineral with a minor quantity of chrysotile that is related to talc-carbonates [48].

## 4. Mechanical and Physical Properties of the Specimens

The physical and mechanical properties of rocks have a crucial influence on their behavior in the application of ornamental uses. In this study, we focused on evaluating different rock samples to determine an index and to classify the most suitable rock types with respect to a combination of their properties. Accordingly, ten samples of every type of rock were examined. Collectively, 100 samples were examined, and the following properties were measured: uniaxial compressive strength (US), slack durability (SDI), point load test

(PLT), water absorption (W), density (D), abrasion (Abr.), and porosity. Table 1 presents the results of the physical and mechanical properties of the specimens.

**Table 1.** Mechanical and physical properties of various rocks.

| Rock Type | Abbreviation | CS Kg/cm$^2$ | SD Kg/cm$^2$ | PLT Kg/cm$^2$ | Water Absorption % | Density g/cm$^3$ | Abrasion g/cm$^2$ | Porosity, % |
|---|---|---|---|---|---|---|---|---|
| Black marble1 | B1 | 387.6 | 89.7 | 70.2 | 0.35 | 2.65 | 0.46 | 0.98 |
| Black marble2 | B2 | 445.7 | 86.4 | 45.6 | 0.39 | 2.78 | 0.51 | 0.3 |
| Black marble3 | B3 | 410.6 | 84.3 | 43.8 | 0.51 | 2.71 | 0.42 | 1.01 |
| White marble1 | W1 | 897.4 | 87.8 | 47.0 | 0.03 | 2.46 | 0.06 | 0.45 |
| White marble2 | W2 | 789.3 | 89.7 | 75.6 | 0.05 | 2.55 | 0.07 | 0.71 |
| Sunny marble | SM | 453.2 | 91.3 | 73.5 | 0.06 | 2.71 | 0.09 | 0.61 |
| Gray granite | GG | 546.8 | 95.4 | 68.9 | 0.03 | 2.67 | 0.31 | 0.42 |
| Red granite | RG | 645.9 | 94.6 | 43.9 | 0.04 | 2.87 | 0.33 | 0.32 |
| Pink granite | PG | 500.8 | 96.5 | 41.8 | 0.05 | 2.75 | 0.21 | 0.29 |
| Serpentine | S | 521.3 | 94.8 | 46.7 | 0.06 | 2.69 | 0.08 | 0.34 |

The specimens were collected from the studied areas and prepared for the test according to the ASTM standard code [49]. The criteria were categorized as physical (water absorption, density, and porosity) and mechanical (compressive strength, slack durability, point load test, and abrasion) properties. All tests were examined withinside the Faculty of Engineering Qena, Al-Azhar University, Egypt. The results of the properties were used as the criteria for calculating the weight for adopting the AHP method, scoring the normalization for every criterion. TOPSIS was used for estimating the geometric distance among every alternative and the ideal alternative in order to illustrate the rank of the exceptional selection.

## 5. AHP: An Approach for Estimating the Weight of the Samples

Numerous papers have studied the use of AHP with regard to mining engineering troubles in general [50]. Ataei et al. [51] carried out a survey among 17 experts who were concerned with mine planning and format process. Musingwini and Minnitt [27] used AHP to rank numerous mining methods practiced within the platinum fields of the Bush-veld Complex in South Africa in the order of efficiency. Balt [52] recognized a need for a realistic method in the mining enterprise to assist engineers to carry out AHP in any discipline in which a choice must be made between multiple alternatives. Based on a couple of selection criteria, Kluge and Malan [53] investigated the utility of AHP for mining engineering troubles. The review demonstrates that AHP is a suitable decision-making device for mining applications [54]. Guo, Q. et al. [55] hooked up an AHP–TOPSIS complete choice version to offer a reference for the optimization of mining techniques for lightly inclined and gently damaged complicated ore bodies at home and abroad. During the early stages of feasibility studies, its simplicity with a qualification of uncertainty was higher than its complexity, which will always increase the level of uncertainty in the face of sparse data. TOPSIS is likewise a recommended approach for choice-making concerning the optimum alternative of decorative natural stones primarily based on their mechanical characteristics, physical characteristics, and cost.

## 6. TOPSIS: Relevance

TOPSIS has been broadly employed to challenge multiple-attribute decision-making problems because of the benefits of the ease of calculation, the resilience of application compared to other methods, and acceptable results [56,57]. Similar to TOPSIS in perfor-

mance is the VIKOR method, which was announced academically in 1998 by Opricovic. However, the VIKOR method has a major downside, which is searching for the compromise ranking order, such as the compromise between the pessimistic and expected solution [58]. On the contrary, scholars were satisfied with the TOPSIS method; thus, they developed it for different fields. Based on TOPSIS, Kaveh et al. transformed the multi-objective decision-making process to a bi-objective decision-making process. Additionally, they offered a framework, which included tangible and intangible determinants, to conclude an integrated multi-objective framework [59]. The study reviews the interest of some scholars in TOPSIS and the use of TOPSIS in different areas.

To solve urban planning problems, Sharma and Singhal applied fuzzy TOPSIS [60]. In concrete production, siliceous materials were assessed by the fuzzy TOPSIS method to extend the concrete's lifespan and save costs to achieve sustainable development in the building sector [61,62]. Based on the possibility theory, Ye and Li [63] used fuzzy TOPSIS based on fuzzy numbers to challenge multi-attribute decision-making. Baykasglu and Golcuk [64] used fuzzy TOPSIS interrelated to fuzzy cognitive maps to solve complicated decision-making problems. Maldonado-Macias et al. utilized an intuitionistic fuzzy TOPSIS to assess cutting-edge manufacturing technologies via numerical control on milling machines [65]. Luis P. and others [66] developed a hesitant fuzzy linguistic term with TOPSIS to evaluate lean manufacturing and to offer an optimal alternative to decision makers. Jin Cheng and others proposed a heterogeneous TOPSIS to solve heterogeneous multi-attribute decision-making problems with deviation and fixation attributes. This proposal successfully used structural optimization of the high-speed press while considering multi-source uncertainties [67]. A hybrid fuzzy analysis network process based on TOPSIS was used to select energy plant locations related to solid waste [68]. TOPSIS was improved to assess power quality based on the correlation between indices, which were ignored by other traditional methods, such as AHP and entropy weight (EW) [69]. Therefore, in this study, we combined AHP and TOPSIS to obtain the optimum decision on the more suitable natural ornamental stone among other samples used in this study.

## 7. Methodology

The method of evaluation of the ten types of natural Egyptian stones involves four steps. The first step is developing a hierarchical structure by determining alternatives and the criteria and structuring the decision hierarchy. The second step is used to assign weights to the parameters by using the AHP technique. The third step is to analyze the assigned weights and ratings of the physical and mechanical parameters by using the TOPSIS technique. The final step is to select the best suitable stones (granite, . . . etc.) for indoor and outdoor uses. This methodology must be used to acquire the most accuracy and smooth application. Figure 3 illustrates the proposed methodology in this study.

### 7.1. Assigning Weight and Rate for Each Criterion

AHP includes decomposing a multi-stage hierarchical structure of objectives, criteria, and alternatives. The decomposition right into a hierarchy is primarily based on preceding studies, research, and empirical experiences. Once the hierarchy is developed, the relative significance of the decision criteria are assessed, after which the decision alternatives regarding every criterion are compared. Finally, the general precedence of every decision alternative and the general rating of the decision alternatives are determined. The evaluation of the relative significance of the decision criteria and the evaluation of decision alternatives regarding every criterion are accomplished by a pair-wise comparison, which concerned the subsequent 3 tasks:

(1) Growing a comparison matrix at every stage of the hierarchy beginning from the second stage and going down.
(2) Computing the relative weights for every detail of the hierarchy.

(3)　Estimating the consistency ratio to test the consistency of the judgment. Table 2 illustrates the weight of the criteria of the properties. Table 3 affords the rate of each criterion. The most significant interval has a score of one and the least has a score of 0.2.

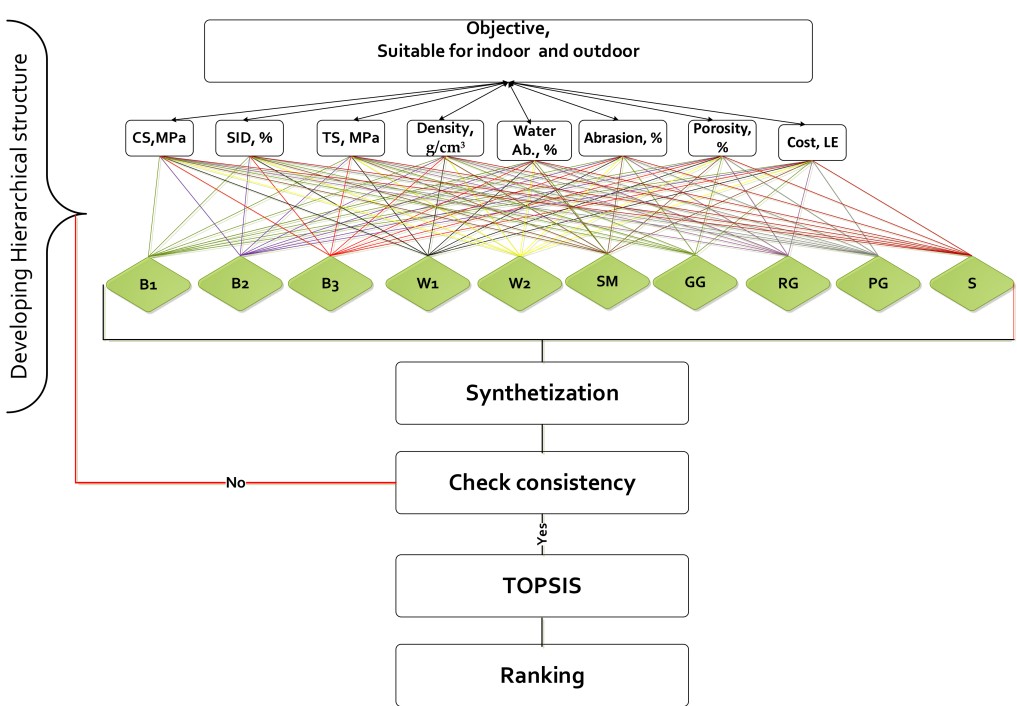

**Figure 3.** Proposed methodology in this study.

**Table 2.** Weight of the criteria of properties.

| Weights Properties | 0.176 | 0.116 | 0.176 | 0.057 | 0.063 | 0.049 | 0.041 | 0.319 |
|---|---|---|---|---|---|---|---|---|
| **Properties** | CS | SD | PLT | W. ab. | D. | Abr. | PO | Cost |

**Table 3.** Assignment of ratings to different parameters [70].

| Parameter | Rating | | | | |
|---|---|---|---|---|---|
| | **0.2** | **0.4** | **0.6** | **0.8** | **1** |
| CS, Kg/cm$^2$ | <250 Very low | 250–500 low | 500–750 Medium | 750–1000 High | >1000 Very high |
| SD, Kg/cm$^2$ | <25 Very low | 25–50 low | 50–100 Medium | 100–150 High | >150 Very high |
| PLT Kg/cm$^2$ | <50 Very low | 50–60 low | 60–75 Medium | 75–100 High | >100 Very high |
| Water absorption, % | >0.75 Very high | 0.75–0.5 high | 0.5–0.25 Medium | 0.25–0.1 low | <0.1 Very low |
| Density, g/cm$^3$ | <2 Very low | 2–2.3 low | 2.3–2.6 Medium | 2.6–2.9 High | >3 Very high |
| Abrasion, % | >0.5 Very high | 0.5–0.3 High | 0.3–0.1 Medium | 0.1–0.05 low | <0.05 Very low |
| Porosity, % | >0.75 Very high | 0.75–0.5 high | 0.5–0.25 Medium | 0.25–0.1 low | <0.1 Very low |
| Cost, EGP | <500 Very cheap | 500–750 cheap | 750–1000 Moderate | 1000–2000 Expansive | >2000 Extremely Expensive |

### 7.2. Ranking Alternatives Based on the TOPSIS Approach

TOPSIS assumes that there is monotonous criteria growth or lower consistency with normalization as a primary factor. However, an unusual size is taken into consideration in multi-criteria cases. TOPSIS is a powerful technique for evaluating criteria that might take into consideration negative consequences in different aspects. Such consequences offer us extra practical modeling in comparison to the consequences of different methods thinking about the associated alternatives to encompass or exclude alternative solutions. One model used 36 criteria that had been categorized into 6 most important groups by Hartman and Mutmansky [71–73].

The discussion and example of the AHP technique in the previous section is clarified earlier than the adjustments are mentioned in this section. The primary criterion for the TOPSIS approach is measuring the stratification of all the mechanical and different properties of the bedrock such that they may be brought to a brand new Excel sheet and connected with the phenomena similar to all mining methods using TOPSIS, which could without difficulty connect all of the characterizations of rock types (e.g., mechanical properties, such as UCS, TS, and abrasion; physical properties, such as density, water absorption, wave velocity; economical elements associated with cost). All equations are formulated and connected using cells in an Excel sheet with all the properties; if the problem follows the alternative, this denotes the cost assigned to the jth criterion of the ith alternative, where xij is the selection matrix. The equivalent weight of the property is expressed as w1, w2, . . . , wn, and the TOPSIS procedure is expressed in 5 steps using Equations (1)–(5).

1.   To normalize the decision matrix:

In this step the decision matrix is converted to normalized decision matrix so that the scores obtained in different scales becomes comparable.

$$\overline{X}_{ij} = \frac{X_{ij}}{\sqrt{\hat{a}_{i=1}^{n} X_{ij}^2}} \tag{1}$$

$r_{ij} = x_{ij} \_\_m_k = 1 \times 2$ $kj$, where $i = 1,...,$ m; $j = 1,...,$ n. $r_{ij}$ denotes the normalized value of the $j^{th}$ criterion for the $i^{th}$ alternative a$i$.

2.   To calculate the weighted normalized decision matrix:

The weighted normalized matrix is obtained by multiplying each column of the normalized decision matrix with the associated criteria weight corresponding to that column. Hence an element vij of weighted normalized matrix V is represented as follows:

$$V_{ij} = \overline{X}_{ij} x \ W_j \tag{2}$$

$V_{ij} = w_j \ r_{ij}$, $i = 1,...,$ m; $j = 1,...,$ n (2), where $w_j$ is the weight of the $j^{th}$ criterion or attribute.

3.   To determine the positive and negative ideal solutions:

This step produces the positive ideal solution ($S_i^+$) and negative ideal solution ($S_i^-$) in the following manner.

$$S_i^+ = \left[ \hat{a}_{j=1}^{m} \left( V_{ij} - V_j^+ \right)^2 \right]^{0.5} \tag{3}$$

4.   To calculate the Euclidean distance from the ideal worst condition

$$S_i^- = \left[ \hat{a}_{j=1}^{m} \left( V_{ij} - V_j^- \right)^2 \right]^{0.5} \tag{4}$$

5.   To calculate the performance score and ranking:

In this step the relative closeness (P*i*) value of each alternative with respect to the ideal solution is determined using the Equation (5). The value of (P*i*) lies within the range from 0 to 1.

$$P_i = \frac{S_i^-}{S_i^+ + S_i^-} \tag{5}$$

## 8. Results and Discussion

### 8.1. Results

Table 4 provides the transformation of the method criteria from AHP methods, wherein all properties are weighed using the AHP method and rated about 1 [74]. Table 5 offers the calculation normalized matrix, as per Equation (1). Table 6 summarizes the outcomes of variables elevated by the weighted index for each property. Table 7 illustrates the positive and negative ideal solutions, and Table 8 provides the final outcomes according to the Euclidean distance from the suitable worst and ranking.

**Table 4.** Criteria for conversion to the new technique according to weight and rate.

| Weights/Rates | 0.176 | 0.116 | 0.176 | 0.057 | 0.063 | 0.049 | 0.041 | 0.319 |
| | CS | SD | PLT | W. ab. | D. | Abr. | PO | Cost |
|---|---|---|---|---|---|---|---|---|
| B1 | 0.4 | 0.6 | 0.6 | 0.2 | 0.8 | 0.4 | 0.2 | 0.2 |
| B2 | 0.4 | 0.6 | 0.2 | 0.2 | 0.8 | 0.2 | 0.6 | 0.2 |
| B3 | 0.4 | 0.6 | 0.2 | 0.2 | 0.8 | 0.4 | 0.2 | 0.2 |
| W1 | 0.8 | 0.6 | 0.2 | 1 | 0.6 | 0.8 | 0.4 | 0.4 |
| W2 | 0.8 | 0.6 | 0.6 | 1 | 0.6 | 0.8 | 0.4 | 0.4 |
| SM | 0.4 | 0.6 | 0.6 | 1 | 0.8 | 0.8 | 0.4 | 0.6 |
| GG | 0.6 | 0.6 | 0.6 | 1 | 0.8 | 0.6 | 0.6 | 0.8 |
| RG | 0.6 | 0.6 | 0.2 | 1 | 0.8 | 0.6 | 0.6 | 0.8 |
| PG | 0.6 | 0.6 | 0.2 | 1 | 0.8 | 0.6 | 0.6 | 1 |
| S | 0.6 | 0.6 | 0.2 | 1 | 0.8 | 0.8 | 0.6 | 1 |

**Table 5.** Normalized matrix.

| | CS | SD | PLT | W. ab. | D. | Abr. | Po | Cost |
|---|---|---|---|---|---|---|---|---|
| B1 | 0.2182 | 0.3162 | 0.4629 | 0.0750 | 0.3310 | 0.2000 | 0.1302 | 0.099 |
| B2 | 0.2182 | 0.3162 | 0.1543 | 0.0750 | 0.3310 | 0.1000 | 0.3906 | 0.099 |
| B3 | 0.2182 | 0.3162 | 0.1543 | 0.0750 | 0.3310 | 0.2000 | 0.1302 | 0.099 |
| W1 | 0.4364 | 0.3162 | 0.1543 | 0.3748 | 0.2483 | 0.4000 | 0.2604 | 0.198 |
| W2 | 0.4364 | 0.3162 | 0.4629 | 0.3748 | 0.2483 | 0.4000 | 0.2604 | 0.198 |
| SM | 0.2182 | 0.3162 | 0.4629 | 0.3748 | 0.3310 | 0.4000 | 0.2604 | 0.297 |
| GG | 0.3441 | 0.3536 | 0.5303 | 0.3769 | 0.3746 | 0.3078 | 0.4286 | 0.3961 |
| RG | 0.3273 | 0.3162 | 0.1543 | 0.3748 | 0.3310 | 0.3000 | 0.3906 | 0.398 |
| PG | 0.3273 | 0.3162 | 0.1543 | 0.3748 | 0.3310 | 0.3000 | 0.3906 | 0.4951 |
| S | 0.3273 | 0.3162 | 0.1543 | 0.3748 | 0.3310 | 0.4000 | 0.3906 | 0.4951 |

**Table 6.** Normalized matrix multiplied through rate for each property.

|  | CS | SD | PLT | W. ab. | D. | Abr. | Po | Cost |
|---|---|---|---|---|---|---|---|---|
| B1 | 0.0384 | 0.0367 | 0.0815 | 0.0043 | 0.0209 | 0.0098 | 0.0053 | 0.0316 |
| B2 | 0.0384 | 0.0367 | 0.0272 | 0.0043 | 0.0209 | 0.0049 | 0.0160 | 0.0316 |
| B3 | 0.0384 | 0.0367 | 0.0272 | 0.0043 | 0.0209 | 0.0098 | 0.0053 | 0.0316 |
| W1 | 0.0768 | 0.0367 | 0.0272 | 0.0214 | 0.0156 | 0.0196 | 0.0107 | 0.0632 |
| W2 | 0.0768 | 0.0367 | 0.0815 | 0.0214 | 0.0156 | 0.0196 | 0.0107 | 0.0632 |
| SM | 0.0384 | 0.0367 | 0.0815 | 0.0214 | 0.0209 | 0.0196 | 0.0107 | 0.0948 |
| GG | 0.0606 | 0.0410 | 0.0933 | 0.0215 | 0.0236 | 0.0151 | 0.0176 | 0.1263 |
| RG | 0.0576 | 0.0367 | 0.0272 | 0.0214 | 0.0209 | 0.0147 | 0.0160 | 0.127 |
| PG | 0.0576 | 0.0367 | 0.0272 | 0.0214 | 0.0209 | 0.0147 | 0.0160 | 0.1579 |
| S | 0.0576 | 0.0367 | 0.0272 | 0.0214 | 0.0209 | 0.0196 | 0.0160 | 0.1579 |

**Table 7.** Positive and negative ideal solutions.

| $V^+$ | 0.0384 | 0.0367 | 0.0272 | 0.0043 | 0.0156 | 0.0049 | 0.0053 |
|---|---|---|---|---|---|---|---|
| $V^-$ | 0.0768 | 0.0410 | 0.0933 | 0.0215 | 0.0236 | 0.0196 | 0.0176 |

**Table 8.** Euclidean distance from the ideal worst and ranking.

| $Si^+$ | $Si^-$ | $Pi$ | Rank |  |
|---|---|---|---|---|
| 0.0548 | 0.1347 | 0.71 | 2 | B1 |
| 0.1495 | 0.1495 | 0.50 | 7 | B2 |
| 0.1496 | 0.1496 | 0.50 | 7 | B3 |
| 0.1161 | 0.1431 | 0.55 | 6 | W1 |
| 0.0962 | 0.0164 | 0.15 | 8 | W2 |
| 0.0754 | 0.1329 | 0.64 | 5 | SM |
| 0.0358 | 0.1275 | 0.78 | 1 | GG |
| 0.0759 | 0.1457 | 0.66 | 4 | RG |
| 0.0693 | 0.1441 | 0.68 | 3 | PG |
| 0.0691 | 0.1440 | 0.68 | 3 | S |

*8.2. Discussion*

After checking the consistency ratio to test the consistency of the judgment, our study constructed the decision matrix and determined the weight of criteria using the AHP method as shown in Table 4. In terms of the values of the weights, the cost is the highest criteria with 0.319, and porosity with the lowest value equals 0.041. Table 5 illustrates that, several attribute dimensions were converted into non-dimensional characteristics, which allowed for cross-criteria comparisons. The scores in the assessment matrix must be changed to a normalized scale because different criteria are frequently measured in different units. One of the various well-known standardized formulas can be used to normalize values according to Equation (1). Table 6 demonstrates the calculations of weighted normalized value Vij. Table 7 presents the semi-final step to determine the positive ideal alternative and the negative ideal alternative (according to Equation (2), $V^+$ and $V^-$). The optimal positive solution maximizes the benefit criteria while minimizing the cost criteria, whereas the ideal negative solution maximizes the cost criteria while minimizing the benefit criteria.

Ideal values S$i^+$, S$i^-$, and performance score P$i$ were calculated related to Equations (3)–(5) respectively. All results are presented with the final ranking in Table 8. It is clearly noted that the gray granite is the highest ranked (1) with a Pi equal to 0.78. Thus, the GG is more suitable for indoor and outdoor uses due to its image of hardness, strength, and durability; granite is an extremely good preference for excessive traffic regions wherein magnificence and fashion are desired. Granite feels at home in a rustic farmhouse in addition to a cutting edge expensive high-rise building. The style of hues and textures are developments that set granite aside from the rest. This awesome stone is good for kitchen countertops, accessory islands, bar tops, dining tables, flooring, etc. The second ranked stone is Black marble (B1) with a Pi equal to 0.71, which makes it appropriate in particular for enclosed spaces. However, marble is a good deal more richly colored and patterned than granite. Marble's splendor will remain for generations and is flexible enough to be used in any part of the house, in such locations as fireplace surrounds, decorative furnishings, walls, flooring, and bathrooms. Marble specifically stands proud within the bath. It may be implemented on nearly every surface, which include vanities, bathroom walls, bathtub decks, and flooring. Marble is more vulnerable to staining through many foods, spilled liquids, and different family substances, and is now no longer advocated for use as kitchen countertops. Softer and more porous than granite, marble is more appropriate for much less trafficked, formal regions.

The third ranked stones are serpentine (S) and pink granite (PG) with a Pi equal 0.68; because of the great durability of serpentine, it is more suitable for both indoor and outdoor application. Serpentine is immune to weathering; however, it suffers from the use of acidic cleaners in indoor use, while the serpentine with an excessive content material of talc used on the outside faces might go through an increase in volume and consequent speedy degradation. On the other hand, pink granite is a very versatile stone and can be used for floors, walls, sinks, countertops, tables, stairs, and more, in relation to its strength and durability. The pink granite is one of the hardest and most resistant granites, withstanding abrasion, heat, water, and scratches. Moreover, no major maintenance is needed to preserve its beauty. However, it is a cold material, ideal for warm places, but in colder regions, it can cool the environment even more. Additionally, because it is a heavier stone, it is necessary to have a firmer base. The pink granites are not very homogeneous stones, which can be a negative point for indoor and outdoor decorations.

The lowest choice stone is White Marble (WM) with a Pi equal to 0.15, which is considered the most expensive stone for indoor and outdoor applications. White Marble has many disadvantages such as being a more costly stone than others, and needing high maintenance for countertops, flooring, and other design applications Due to marble being a porous and soft stone, it is vulnerable to staining, chipping, and scratches. A chart summarizing the ranks of stones is shown in Figure 4.

AHP and TOPSIS have inherent flaws that can be overcome by combining the two techniques. AHP can be used to calculate weights for the weight elicitation problem within TOPSIS. TOPSIS can compensate for the information processing limitation and time-consuming pair-wise comparison procedure for AHP. In order to combine the benefits of AHP and TOPSIS while overcoming their individual shortcomings, a mixed AHP–TOPSIS model is proposed here, in which AHP techniques are used to calculate relative importance criteria weights and TOPSIS procedures are used to calculate final rankings. Therefore, the AHP–TOPSIS mixed model is best suited for predicting selection in bulk commodities or high-priced product categories where choice accuracy is critical (as tiny weight elicitation differences can lead to huge discrepancies in the final results).

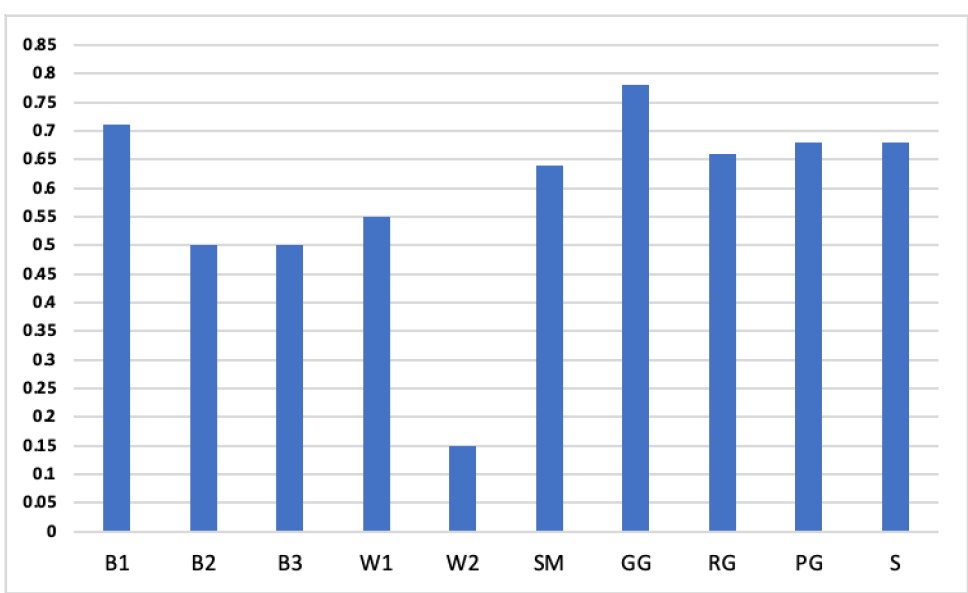

**Figure 4.** Ranking stones according to the TOPSIS technique.

The combined AHP–TOPSIS tackles the deficiencies of some MCDM techniques analytically and comprehensively. The analytical procedures help decision makers break down complex problems into actionable tasks, extending the model's relevance to additional decision-making scenarios and simplifying information input processes. The AHP–TOPSIS model has no restrictions on attributes or alternative numbers in general. It can deal with a diverse set of importance weights, attributes, alternatives, and decision makers. AHP–TOPSIS integration has a wide range of applications. Examples include defining ideal management plans [75], material selection in engineering design [76], effectiveness evaluation for manufacturing organizations [77], provider segment [78], consumer industrial design processes [79], and mined land suitability analysis [80]. According to Tavana and Hatami-Marbini [81], the AHP–TOPSIS mixed model assists decision makers in three ways: (a) breaking down complex problems into manageable and hierarchical steps, (b) eliminating the biasness of decision making by verifying consistency ratios within AHP, and (c) achieving final scores through rigorous logical steps embedded within TOPSIS.

According to the literature, the hybrid AHP–TOPSIS approach for selecting ornamental stones for specific purposes has yet to be implemented. As previously stated, (citations 75–81), the majority of applications were found in management, material selection in engineering design, manufacturing companies, problems of supplier selection, consumer-driven product design, mined land suitability analysis, and human spaceflight mission planning. As a result, this is the first study to apply the AHP–TOPSIS method to ornamental stones in order to evaluate their physical and mechanical properties for proper use as indoor and outdoor building finishes.

## 9. Limitations and Implications of the Study

The AHP–TOPSIS mixed model combines the advantages of AHP (which can compare alternatives in pairs to derive weights) and TOPSIS (which does not have capacity limitations on the number of attributes and alternatives). This hybrid approach can thus be used when decision makers are unable to provide weightings for a large number of alternatives or when very precise weights are required.

Because TOPSIS cannot induce weights, one must depend on other weighting methods such as AHP. As a result, if the weights are not precise, using the TOPSIS method may be impractical. TOPSIS, in the same way as AHP, can result in rank reversal, which occurs when more criteria are added/removed, causing alternative preferences to change. It does, however, have the fewest rank reversals among many methods. TOPSIS, on the other hand,

can find the best alternative faster than most MCDM. Thus, one limitation of the study is that only AHP was used to elicit weights. This means that there is still room for other methods capable of eliciting and calculating weights. As a result, a future study could look into other weight elicitation methods and their impact on final rankings.

## 10. Conclusions

The results confirmed that choosing a suitable stone for indoor and outdoor building areas relies upon the specifications of stones and mechanical and monetary considerations. The optimum use of natural stones may be decided by assessing such types. In this study, TOPSIS was modified by specializing in linking all parameters associated with all criteria to obtain correct results.

The results offer indicators to decision makers to choose an appropriate rock type based on the overall factors assigned to all rock properties. Applying TOPSIS after enhancing AHP for construction materials is a unique application, regardless of different studies that have adopted TOPSIS in various applications. Scientifically, as mentioned in Section 1, the excessive creditability of TOPSIS as a technique alongside the flexibility and concise statistics is the inducement behind developing it on this paper.

The grey granite sample had the shortest geometric distance from the positive ideal answer and the longest geometric distance from the negative ideal answer with Pi = 0.78. Black marble was 2nd with a Pi = 0.71, while serpentine and pink granite were the 3rd ranked stones with a Pi = 0.68. The results offer the stakeholders with strategic plans to select from the extraordinary natural decorative stones based on the overall factors assigned to all rock specifications and their costs.

**Author Contributions:** The authors state that this paper has been authored in equal contribution with the following details: Conceptualization, M.A.M.A. and A.M.H.; methodology, A.M.A.S.; software, J.-G.K. and M.A.M.A.; validation, G.R., A.M.H. and A.M.A.; formal analysis, M.A.M.A., W.R.A. and A.M.H.; investigation, A.M.A.S. and A.M.A.; resources, M.A.M.A. and G.R.; data collection, A.M.A.S.; writing, M.A.M.A., A.M.H. and W.R.A.; writing—review and editing, J.-G.K., W.R.A. and G.R.; visualization, A.M.A.; supervision, M.A.M.A.; project administration, A.M.A.S.; funding acquisition, G.R. All authors have read and agreed to the published version of the manuscript.

**Funding:** This research was funded by National Natural Science Foundation of China, grant number 52174087.

**Institutional Review Board Statement:** Not applicable.

**Informed Consent Statement:** Not applicable.

**Data Availability Statement:** Raw data from the study are available on request.

**Acknowledgments:** The authors are very grateful for the support of the fund.

**Conflicts of Interest:** The authors declare no conflict of interest.

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
