# Peer review of "Sustainable Assignment of Egyptian Ornamental Stones for Interior and Exterior Building Finishes Using the AHP-TOPSIS Technique"

_sustainability, doi:10.3390/su14042453_

Round 1

Reviewer 1 Report

The paper reads fine in general. However, there is one key element missing in the manuscript which is the problem statement. Even though the authors clearly states the aim of the manuscript which is also captured in the title, the authors are not very clear about the problem this manuscript intends to solve. 

Moreover, even though the authors have presented some benefits of the approach adopted in the paper, the paper generally lacks a clear theoretical and practical significance which I would encourage the authors to present or make clear.

The introduction reads like a literature review with very limited focus on the key constructs that are supposed to be captured in the introduction. For instance, while I see a brief background and the aim clearly presented, some important sections including what I mentioned earlier (e.g., the problem statement, theoretical and practical significance) of the study is missing in the introduction. I encourage the authors to revise and include these sections.

Methodology: The authors should have presented the application of the equations provided in estimating the results generated. The presentation of the formulas generally makes no significant contribution to the paper as it can be presented as an appendix. However, providing an example of how the formulas are applied to arrive at the results is very significant.

Lastly at the results and discussion section, I see no discussions but the presentation of several tables with no descriptions or explications. That is very poor from the authors. What are the implications of the tables provided. What do they mean. What are the implications to other extant studies, results or approaches. This is perhaps the most substandard section of the paper that needs drastic improvements.

I hope the comments provided help. Thanks.

Author Response

Rebuttal and List of Changes

Dear Prof. Editor-in-chief,
Sustainability: Manuscript ID: sustainability-1572114

Title:  Sustainable Assignment of Egyptian Ornamental Stones for Interior and exterior Building Finishes Using the AHP-TOPSIS Technique

Each referee's contribution to our article is greatly appreciated. As listed below, we improved our manuscript in response to the reviewers' suggestions.

Reviewer #1

Reviewers’ comments

Authors’ rebuttal

Introduction

1. The paper reads fine in general. However, there is one key element missing in the manuscript which is the problem statement. Even though the authors clearly state the aim of the manuscript, which is also captured in the title, the authors are not very clear about the problem this manuscript intends to solve. 

This is a very insightful comment, and we recognize that the focal points of the problem statement have been stated in the abstract (Please have a look at page 1 lines 21 to 23). More information about the problem statement has also been added to the introduction. Please see page 2 for more information. We hope that this addition demonstrates the core issue of our research.

2- Even though the authors have presented some benefits of the approach adopted in the paper, the paper generally lacks a clear theoretical and practical significance which I would encourage the authors to present or make clear.

The introduction reads like a literature review with very limited focus on the key constructs that are supposed to be captured in the introduction. For instance, while I see a brief background and the aim clearly presented, some important sections including what I mentioned earlier (e.g., the problem statement, theoretical and practical significance) of the study is missing in the introduction. I encourage the authors to revise and include these sections.

Well, that is a very good comment. We talked about it in the introduction. Using the AHP-TOPSIS Technique, we focused on the problem and the importance of the study.

Please read the introduction once more. The theoretical and practical significance of the study can be found on page 2 between lines 5 and 44..

2-Methodology: The authors should have presented the application of the equations provided in estimating the results generated. The presentation of the formulas generally makes no significant contribution to the paper as it can be presented as an appendix. However, providing an example of how the formulas are applied to arrive at the results is very significant.

Thank you for your input. In addition, all equations have been illustrated to assist readers in following TOPSIS technique procedures.

This study is geared toward interested scholars, and in order to avoid duplicating data, a brief demonstration for each equation has been provided. This addition, we believe, would improve the clarity of these equations. Please refer to pages 11-12.

5-Lastly at the results and discussion section, I see no discussions but the presentation of several tables with no descriptions or explications. That is very poor from the authors. What are the implications of the tables provided. What do they mean. What are the implications to other extant studies, results or approaches. This is perhaps the most substandard section of the paper that needs drastic improvements.

We fully agree with the reviewer. We took this comment very seriously because it was the most dissatisfactory section of the paper that needed drastic improvements. This section has been completely developed, describing the tables and analyzing data to explain the implications of the tables and make the paper more useful for readers. Please see the discussion section (Pages 14 and 15); the highlighted text follows the reviewer's comment.

Reviewer 2 Report

In this research, some Egyptian ornamental stones were evaluated by combining the analytic hierarchy process (AHP) and technique for order preference by similarity to ideal solution (TOPSIS). The AHP-TOPSIS comprehensive decision model optimization was implemented on natural materials relevant to ornaments and finishing purposes of the indoor and outdoor buildings.

MCDM methods are still uncommon for building materials evaluation and comparison. The article uses ones of the oldest multicriteria methods AHP and TOPSIS. There are many new MCDM methods that are also popular and widely used nowadays. I miss a more detailed overview of the application of MCDM methods used in the evaluation of building materials. The article should provide a more detailed explanation why the new MCDM methods are ignored, meanwhile some of the oldest methods are chosen. Old literature sources are cited. Only seven sources are cited which are published within the last three years. AHP and TOPSIS methods are used. However, their authors Saaty and Hwang and Yoon are not cited. This is incorrect. Names and surnames of authors have been swapped in many places in the bibliography (28,29,31,35, etc.).

Two MCDM methods are used. However, the article presents one of them only(TOPSIS).

I suggest to presen the formulas of the method in a more precise way. TOPSIS is incorrectly compared to VIKOR.

Cite:

Sustainability 13(18)10438, 2021

Algorithms 12(6)119, 2019

Archives of mining sciences 58(1)255-267

Sustainability 12(22)9482, 2020.

Symmetry-Basel 11(3)393, 2019.

Author Response

Rebuttal and List of Changes

Dear Prof. Editor-in-chief,
Sustainability: Manuscript ID: sustainability-1572114

Title:  Sustainable Assignment of Egyptian Ornamental Stones for Interior and exterior Building Finishes Using the AHP-TOPSIS Technique

Each referee's contribution to our article is greatly appreciated. As listed below, we improved our manuscript in response to the reviewers' suggestions.

Reviewer #2

Reviewers’ comments

Authors’ rebuttal

Introduction

1- MCDM methods are still uncommon for building materials evaluation and comparison. The article uses ones of the oldest multicriteria methods AHP and TOPSIS. There are many new MCDM methods that are also popular and widely used nowadays. I miss a more detailed overview of the application of MCDM methods used in the evaluation of building materials.

The article should provide a more detailed explanation why the new MCDM methods are ignored, meanwhile some of the oldest methods are chosen

It is a critical question. Thank you for putting in the effort.

We did, however, discuss this issue prior to writing this manuscript. We anticipate that readers and critics will wonder why TOPSIS and AHP, among other new techniques, were chosen. We argued and highlighted this point in the introduction section. kindly, Take a look at lines 26-44 on page 2.

2-Old literature sources are cited. Only seven sources are cited which are published within the last three years. AHP and TOPSIS methods are used. However, their authors Saaty and Hwang and Yoon are not cited. This is incorrect. Names and surnames of authors have been swapped in many places in the bibliography (28,29,31,35, etc.).

So far, we have rooted literature sources such as Saaty and Hwang, as well as Yoon. We have also updated the references with more recent publications. As a result, the percentage of papers published in the last five years has become noticeable in the references list.

We double-checked all of our references.

Please look over this list. Thank you for your insightful comment.

3. Two MCDM methods are used. However, the article presents one of them only (TOPSIS).

We disagree with the comment that the AHP is used to calculate the weight of each criterion in order to be ready for use in the TOPSIS Technique. As a result, it is a unified approach based on two tools (AHP-TOPSIS Technique).

4- I suggest to present the formulas of the method in a more precise way. TOPSIS is incorrectly compared to VIKOR.

Well, we'd like to draw the attention of our reviewers to the fact that TOPSIS is a well-established method, according to numerous studies. As a result, many academics have used it in their research.

VIKOR, for example, has a significant disadvantage. Stefan Z. and colleagues reported in their paper published in the International Journal of Economics and Law, vol. 8, that "... the greatest disadvantage of VIKOR is searching for the compromise ranking order, such as a compromise between pessimistic and expected solutions." The ranking order changes as the weight of these solutions changes...").

There are also numerous papers that discuss the drawbacks of VIKOR. As a result, we chose MCDM and TOPSIS.

5 Cite:

Sustainability 13(18)10438, 2021

Algorithms 12(6)119, 2019

Archives of mining sciences 58(1)255-267

Sustainability 12(22)9482, 2020.

Symmetry-Basel 11(3)393, 2019.

All of those references have been modified and expanded upon.

Round 2

Reviewer 1 Report

Many thanks to the authors for the changes and the improvements made to the manuscript. 

Most of my comments have been fully addressed by the authors. However, I believe the authors can make the final corrections to the manuscript to further enhance its quality. Please, kindly find my comments below:

  1. There is no need for the authors to enumerate/number the various paragraphs at the introduction section. Just presenting them as normal sentence paragraphs should be fine.
  2. Glad to see the development of the new discussion section. I would further encourage the authors to further expand and enhance the quality of the discussions by cross-referencing the results to that of other extant studies. For instance, other than the explications or the elaborations of the tables (i.e., 4-8), there is not a single reference cited in the discussions. It makes it difficult to compare your results with other relevant studies in order to see how your study originally contributes to the scholarship of exterior building finishes and other relevant themes captured in your studies.
  3. New sections on limitations and the study's implications can be presented prior to the conclusions of the study.
  4. Lastly, I would encourage the authors to do a final proofreading of the manuscript.

Thanks very much and wishing you the very best.

Author Response

Rebuttal and List of Changes

Dear Prof. Editor-in-chief,
Sustainability: Manuscript ID: sustainability-1572114

Title:  Sustainable Assignment of Egyptian Ornamental Stones for Interior and exterior Building Finishes Using the AHP-TOPSIS Technique

Each referee's contribution to our article is greatly appreciated. As listed below, we improved our manuscript in response to the reviewers' suggestions.

Reviewer #1

Reviewers’ comments

Authors’ rebuttal

Many thanks to the authors for the changes and the improvements made to the manuscript. 

Most of my comments have been fully addressed by the authors. However, I believe the authors can make the final corrections to the manuscript to further enhance its quality. Please, kindly find my comments below:

  1. There is no need for the authors to enumerate/number the various paragraphs at the introduction section. Just presenting them as normal sentence paragraphs should be fine.

  1. Glad to see the development of the new discussion section. I would further encourage the authors to further expand and enhance the quality of the discussions by cross-referencing the results to that of other extant studies. For instance, other than the explications or the elaborations of the tables (i.e., 4-8), there is not a single reference cited in the discussions. It makes it difficult to compare your results with other relevant studies in order to see how your study originally contributes to the scholarship of exterior building finishes and other relevant themes captured in your studies.

  1. New sections on limitations and the study's implications can be presented prior to the conclusions of the study.

  1. Lastly, I would encourage the authors to do a final proofreading of the manuscript.

Thanks very much and wishing you the very best.

We appreciate the previous comments rising by the reviewer. They were very valuable and enriched our manuscript in this round.

1.       It sounds better to obey the reviewer's comment. Please see section 1. Introduction, page 2. We have modified the introduction and muted numbering of the various paragraphs at the introduction section.

2.        The following paragraphs have been added to section 8.2. discussion (pages 15 & 16). Some references have been cited throughout this section to add creditability. Thank you once again for this critical comment.

3.        

4.        AHP and TOPSIS have inherent flaws that can be overcome by combining the two techniques. AHP can be used to calculate weights for the weight elicitation problem within TOPSIS. TOPSIS can compensate for the information processing limitation and time consuming pair-wise comparison procedure for AHP. In order to combine the benefits of AHP and TOPSIS while overcoming their individual shortcomings, a mixed AHP-TOPSIS model is proposed here, in which AHP techniques are used to calculate relative importance criteria weights and TOPSIS procedures are used to calculate final rankings. Therefore, the AHP-TOPSIS mixed model is best suited for predicting selection in bulk commodity or high-priced product categories where choice accuracy is critical (as tiny weight elicitation differences can lead to huge discrepancies in the final results).

The combined AHP-TOPSIS tackles the deficiencies of some MCDM techniques analytically and comprehensively. The analytical procedures would help decision-makers break down complex problems into actionable tasks, extending the model's relevance to additional decision-making scenarios and simplifying information input processes. AHP-TOPSIS model has no restrictions on attribute or alternative numbers in general. It can deal with a diverse set of importance weights, attributes, alternatives, and decision makers. AHP-TOPSIS integration has a wide range of applications. Examples include defining ideal management plans [75], material selection in engineering design [76], effectiveness evaluation for manufacturing organizations [77], provider segment [78], consumer industrial design processes [79], and mined land suitability analysis [80]. According to Tavana and Hatami-Marbini [81], the AHP-TOPSIS mixed model assists decision makers in three ways: (a) breaking down complex problems into manageable and hierarchical steps, (b) eliminating the biasness of decision-making by verifying consistency ratios within AHP, and (c) achieving final scores through a rigorous logical steps embedded within TOPSIS.

According to the literature, the hybrid AHP-TOPSIS approach for selecting ornamental stones for specific purposes has yet to be implemented. As previously stated (Citations 75-81), the majority of applications were found in management, material selection in engineering design, manufacturing companies, problems of supplier selection, consumer-driven product design, mined land suitability analysis, and human spaceflight mission planning. As a result, this is the first study to attempt the AHP-TOPSIS method to ornamental stones in order to evaluate their physical and mechanical properties for proper use as indoor and outdoor building finishes.

3. We have added a new section. Kindly, have a look at section 9 (pages 16-17) which includes briefly the following:

9.       Limitations and implications of the study

The AHP-TOPSIS mixed model combines the advantages of AHP (which can compare alternatives in pairs to derive weights) and TOPSIS (which does not have capacity limitations on the number of attributes and alternatives). This hybrid approach can thus be used when decision makers are unable to provide weightings for a large number of alternatives or when very precise weights are required.         

Because TOPSIS cannot induce weights, one must depend on other weighting methods such as AHP. As a result, if the weights are not precise, using the TOPSIS method may be impractical. TOPSIS, like AHP, can result in rank reversal, which occurs when more criteria are added/removed, causing alternative preferences to change. It does, however, have the fewest rank reversals among many methods. TOPSIS, on the other hand, can find the best alternative faster than most MCDM. Thus, one limitation of the study is that only AHP was used to elicit weights. This means that there is still room for other methods capable of eliciting and calculating weights. As a result, a future study could look into other weight elicitation methods and their impact on final rankings.

4.We have done a proofread on the manuscript. 

We really appreciate the reviewer's comments which contributed to develop the manuscript in its second round and make it more readable.

Reviewer 2 Report

Citation sources must be provided by all authors
surnames

Author Response

Rebuttal and List of Changes

Dear Prof. Editor-in-chief,
Sustainability: Manuscript ID: sustainability-1572114

Title:  Sustainable Assignment of Egyptian Ornamental Stones for Interior and exterior Building Finishes Using the AHP-TOPSIS Technique

Each referee's contribution to our article is greatly appreciated. As listed below, we improved our manuscript in response to the reviewers' suggestions.

Reviewer #2

Reviewers’ comments

Authors’ rebuttal

Citation sources must be provided by all authors
surnames

Well, the references list has been modified with authors' surnames. Kindly, check our changes in the tail of manuscript.